# Nanometer scale difference in myofilament lattice structure of muscle alters muscle function in a spatially explicit model

**Travis Carver Tune** [1]*, **Simon Sponberg**[1,2]

**1** School of Physics, Georgia Institute of Technology, Atlanta, Georgia, United States of America, **2** School of Biological Sciences, Georgia Institute of Technology, Atlanta, Georgia, United States of America

* ttune3@uw.edu

**Data availability statement:** This model is available at https://github.com/travistune3/multifil_titin/tree/multifil_manduca_workloops.

## Abstract

Crossbridge binding, state transitions, and force in active muscle is dependent on the radial spacing between the myosin-containing thick filament and the actin-containing thin filament in the filament lattice. This radial spacing has been previously shown through spatially explicit modeling and experimental efforts to greatly affect quasi-static, isometric, force production in muscle. It has recently been suggested that this radial spacing might also be able to drive differences in mechanical function, or net work, under dynamic oscillations like those which occur in muscles *in vivo*. However, previous spatially explicit models either had no radial spacing dependence, meaning the radial spacing could not be investigated, or did include radial spacing dependence but could not reproduce *in vivo* net work during dynamic oscillations and only investigated isometric contractions. Here we show the first spatially explicit model to include radial crossbridge dependence which can produce mechanical function similar to real muscle. Using this spatially explicit model of a half sarcomere, we show that when oscillated at strain amplitudes and frequencies like those in the main flight muscles of the hawkmoth *Manduca sexta*, mechanical function (net work) does depend on the lattice spacing. In addition, since the trajectory of lattice spacing changes during dynamic oscillation can vary from organism to organism, we can prescribe a trajectory of lattice spacing changes in the spatially explicit half sarcomere model and investigate the extent to which the time course of lattice spacing changes can affect mechanical function. We simulated a half sarcomere undergoing dynamic oscillations and prescribed the Poisson's ratio of the lattice to be either 0 (constant lattice spacing) or 0.5 (isovolumetric lattice spacing changes). We also simulated net work using lattice spacing data taken from *M. sexta* which has a variable Poisson's ratio. Our simulation results indicate that the lattice spacing can change the mechanical function of muscle, and that in some cases a 1 nm difference can switch the net work of the half sarcomere model from positive (motor-like) to negative (brake-like).

**Funding:** This work was supported by grant W911NF-14-1-0396 from the Army Research Office, National Science Foundation Early Career Development Award MPS/PoLS 1554790, National Science Foundation BII Integrative Movement Sciences Institute 2319710, and the Georgia Tech Dunn Family Professorship to S.S. Other support was provided by the National Science Foundation SAVI student research network in physics of living systems (1205878). The original model was developed with additional support from Army Research office grant W911NF-13-1-0435. T.T. was supported with funding from the University of Washington Center for Translational Muscle Research, and by the NIH National Institute of Arthritis and Musculoskeletal and Skin Diseases under Award Number P30AR074990, and by the NIH grants R01HL157169, R01HL142624, and an American Heart Association Collaborative Sciences Award. The funders had no role in study design, data collection and analysis, decision to publish, or preparation of the manuscript.

**Competing interests:** The authors have declared that no competing interests exist.

## Author summary

The myosin motors which are responsible for force generation in muscle not only produce axial force, but also produce radial force which can deform the myofilament lattice. Previous spatially explicit models investigated how this radial force and lattice spacing might influence isometric force, but were not able to generate net work under dynamic, phasically activated oscillations like those in *in vivo* muscle, known as work loops. Here we revise a previous spatially explicit model and use it to investigate how the structure of the lattice spacing can affect whole muscle mechanical function during simulated work loops.

## 1. Introduction

In muscle, force is generated by the collective action of billions of myosin motors all undergoing nanometer scale conformational changes. However, the mechanical work output of a whole muscle, which is often the physiologically relevant parameter for animal locomotion, happens at the centimeter scale [1]. The multiscale interactions of stress, strain, binding, and activation are challenging but potentially tractable because muscle is a highly ordered, hierarchical tissue [2]. For example, the interactions between chains of sarcomeres can produce emergent history-dependent behavior such as residual force enhancement that single sarcomeres might not [3,4]. While this multiscale interplay has led to perhaps a greater understanding of molecular to macroscopic function in muscle than in any other tissue, it is challenging to extend this mechanistic understanding from quasi-static regimes to the dynamic behavior that makes muscle so versatile during movement. Here, we show in a spatially explicit, half-sarcomere model how the nanometer scale lattice structure of muscle can affect whole muscle mechanical function under dynamic conditions relevant for locomotion.

Tissue-scale physiological properties of whole muscle arise from the underlying 3D structure and geometry of muscle sarcomeres and myofilament lattice. For example, whole muscle's force-length relationship was originally attributed to the amount of overlap between myosin-containing thick filaments and the actin-containing thin filaments at the micron scale [5,6]. However, the radial spacing between the thick and thin filaments is not constant and changes with sarcomere axial strain changes [7]. Not only that, but crossbridges (myosin motors bound to actin) can generate radial forces of comparable strength to axial forces, which in turn can deform the lattice [8–10]. Therefore, the lattice spacing and crossbridge binding are coupled together, influencing each other. Prior spatially explicit model of muscle's contractile lattice, showed that the radial separation of thick and thin filaments can contribute between 20%–50% of the change in force in the quasistatic force-length curve [11,12].

Because these previous modeling and experimental efforts considered lattice spacing in quasistatic conditions, we wondered if this radial separation could significantly affect a whole muscle's mechanical function under dynamic conditions such as those experienced during cyclic locomotion. Since thick and thin filaments are arranged in a highly ordered hexagonal crystal lattice, the thick-thin filament radial spacing can be measured experimentally with time-resolved X-ray diffraction [13–16]. We can now measure lattice spacing force simultaneous with macroscopic measurements of activated muscle's force length curve (a "work loop") under physiologically relevant conditions [17–19]. Previously, we used this approach [20] to explore the differences in two muscles in the cockroach *Blaberus discoidalis*, which have very similar quasistatic properties yet very dissimilar work outputs [21]. We found that the two muscles have a one nanometer difference in their myofilament lattice spacing at rest but very

similar lattice spacing under activated quasistatic conditions. During contraction they therefore have different lattice spacing dynamics. The differences in the force during work loops in the two muscles correlated to lattice spacing differences. This suggested that the nanometer-scale differences in lattice spacing of a muscle could potentially explain the macroscopic whole muscle function [20].

However, it is experimentally hard to show that a lattice spacing change can by itself change the work output of a whole muscle. While chemicals like dextran can be used to increase lattice spacing osmotically, this usually requires removing the cellular membrane ("skinning") [11]. Skinning the muscle makes isolating the effect of lattice spacing on mechanical work of the intact muscle difficult because the sarcolemma provides a stabilizing radial force to the lattice [22,23]. So to test the effect of lattice spacing on muscle mechanical work output independent of other changes, we turned to a spatially explicit three-dimensional model of a muscle half sarcomere [11,24,25]. The fact that the model is spatially explicit means that the model can allow us to investigate how the lattice spacing can affect mechanical work at the sarcomere scale. The model allows us to prescribe not only a fixed radial filament spacing, but any trajectory of changes over a strain cycle. This is relevant since while in some muscles the lattice spacing is approximately constant with length change [26], in other muscles the lattice spacing depends strongly on length. This relationship can be characterized by the Poisson ratio, $\nu = \frac{d_{radial}}{d_{axial}}$, the ratio between strain changes in the radial and axial directions.

To provide the first spatially explicit models testing the effects of lattice spacing on dynamic muscle function we first adapt previous models to produce reasonable work loops in a physiologically accurate range. As with previous spatially explicit muscle models, we fixed parameters based on insect skeletal muscle taken from the dorsolongitudinal muscle (DLM) of the hawkmoth, *Manduca sexta*, the muscle responsible for the downstroke of the wings during flight [27–29]. We chose this muscle specifically because the mechanics of it have been studied, but because detailed x-ray diffraction measurements of its structure exist [30]. We validate the model by comparing it to twitch and tetanus force responses as well as mechanical work at different phases of activation. We then simulate net (axial) mechanical work under different lattice spacing offsets and trajectories to test if the lattice spacing changes on the scale of a single nanometer can modulate mechanical work, consistent with what was observed in the two cockroach muscles. We test the different lattice spacing dynamics around these offsets (constant, isovolumetric, and experimentally derived) to generalize the results. Finally, we incorporate a recent model of titin (and insect titin-like molecules) to test if the effects of lattice spacing are influence by these molecules which may significantly influence work production especially under dynamic conditions.

## 2. Materials and methods

Earlier versions of the spatially explicit models explored work production under periodic contractions, but did not model any effects of the lattice (i.e. radial) spacing, which meant the effect of lattice spacing could not be investigated [29]. A later version of the model did include explicit radial spacing [12,31]. However, while this more recent model was able to produce good quasi-static results, which was the goal of those studies, it was unable to produce physiologically realistic amounts of mechanical work during high frequency, high strain, oscillations. Here, we describe the model as well as the modifications we made.

## 2.1. Model overview

Our basis for the model is taken from [12,31,32]. Each time step in the model follows a sequence of steps that ultimately give a scalable estimate of axial force produced by the myofilament lattice. Starting at the initial spatial configuration of the model, each myosin head first undergoes thermal forcing by drawing energies from a Boltzmann distribution for each spring that comprises the myosin head, which is then used to update the position of the heads. Then binding probabilities for each myosin head are calculated for the new spatial configuration of the half sarcomere and a set of prescribed rate equations (see below). After transitions between the states have been performed, the nodes which make up the thick and thin filaments undergo a minimization procedure to find the equilibrium configuration of the half sarcomere. This loop of diffusion, stochastic transition, and then force balancing is repeated at each time step.

Earlier versions of the spatially explicit model (termed 2sXB to indicate the torsional 2 spring system which makes up the crossbridge head, as opposed to a single linear spring in the axial direction) investigated isometric muscle's force-length dependence on actin-myosin spacing [12,31]. Those models were able to capture muscle's quasi-static behavior and to show that the force-length relationship in muscle is in fact highly dependent on radial spacing changes of actin and myosin which are coupled to changes in sarcomere axial length [12]. This is what led us to use that model to investigate if the actin-myosin spacing could have a significant effect on net work of a sarcomere.

The net (mass-specific) mechanical work of muscle is given by the area enclosed by a stress-strain curve in which the muscle is periodically activated, called the muscle's work loop [1,18,33]. In work loop experiments, typically the *in vivo* strain amplitude, frequency, and pattern of activation for a given muscle during a given behavior are measured in an intact animal, allowing the same patterns to be input into an excised muscle, from which net work can be measured [21]. After establishing the behavior of the muscle under conditions which mimic its *in vivo* behavior, the parameters of the work loop can be adjusted to explore the properties of muscle [17]. For example, the phase of activation - the timing of activation relative to the strain cycle - can be adjusted, yielding a phase sweep. While the *in vivo* range of phase of activation might be limited, by expanding the range of activation in work loops we can drive the muscle into different force producing regimes to examine its function.

While ideal for capturing axial and radial force contributions, the prior models could not produce significant positive work under *in vivo* frequencies and amplitudes. We simulated work loops using the release version of these models at 25 Hz at 10 phases of activation between 0 and 0.9 and compared the results to phase sweep work loop data taken from *M. sexta* isolated, whole muscle experiments [27]. We found that work loops produced orders of magnitude more net negative work (-230 J kg$^{-1}$ at phase of activation of 0 in simulation compared to 2 J kg$^{-1}$ in real muscle) under these conditions (Fig 1). It is important to acknowledge that this dynamic regime with high rates of axial shortening and lengthening were not the purpose of the prior model and these simulations only serve to illustrate the regime where modifications are necessary to apply such approaches. Other prior models that did not include a second spring, and hence an explicit radial dependency, could emulate work production under cyclic stress-strain curves, but cannot test the dependency on lattice spacing [29]. Here, we describe the model geometry and adaptations that were made to extend the model's dynamic range to *in vivo* strain frequencies and amplitudes.

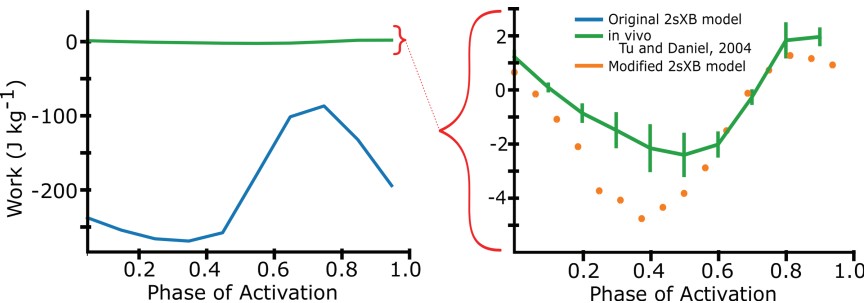

**Fig 1. Net Work vs. phase of activation for prior and modified model.** Here we show the net work of the previous spatially explicit model (blue) compared to *M. sexta* for different phases of activation (green). Work loop simulations were done at 25 Hz and 10% peak-to-peak amplitude at a sarcomere length of 2.5 $\mu$ m, which is the *in vivo* frequency and amplitude of *M. sexta*. As an inset, we show simulations after our modifications to the spatially explicit model presented in the paper. *M. sexta* work loop data first published in [27].

## 2.2. Model geometry

As in [12], a half sarcomere is represented as a 3 dimensional spring lattice. Myosin-containing thick and actin-containing thin filaments are composed of a series of linear springs (Fig 2) where nodes between springs represent either the origin of a myosin motor (in the case of the thick filament) or a potential binding site (in the case of the thin filament). The model consists of 4 thick filaments and 8 thin filaments arranged such that one thin filament is located equidistant between three thick filaments, as in vertebrate muscle [9]. Each thick filament is attached to the z-disc by titin, which attaches to the z-disk and to the thin filament. This spatially explicit unit (Fig 2) is the repeating motif that composes the regular myofilament lattice in a sarcomere. Periodic boundary conditions are enforced so that each thick filament interacts with 6 thin filaments and allow us to scale to arbitrary size. Interactions with the boundary of sarcomere and fluid interaction within the sarcomere are currently ignored.

Each node of the thick filaments contains triplets of myosin heads, referred to as crowns. The elastic links between adjacent crowns are described as linear springs with a set length of 14.3 nm, consistent with the 14.3 nm repeat in muscle which gives the helical repeat of the myosin heads [34]. Myosin head triplets are azimuthally distributed by 120deg and adjacent crowns are rotated in a pattern of 60deg, 60deg, 0deg , as found in [35]. Thin filaments are similarly composed of crossbridge binding sites which are spaced 38.7 nm apart and are linked together by linear springs. As crossbridges bind, filament strain can change the local spacing of heads or binding sites and can arise from either muscle stretch or internal, local, axial stress produced from myosin binding. The out-of-register nature of myosin heads and binding sites (42.9 nm vs 38.7 nm) is a well-known feature of muscle that emphasizes the importance of a spatially explicit model because compliance in the filaments can either promote or suppress binding probability [13,24].

| | Thick filament | Thin filament |
|---|---|---|
| stiffness | $2020\ pN \cdot nm^{-1}$ | $1760\ pN \cdot nm^{-1}$ |
| $l_0$ | $14.3\ nm$ | $12.2\ nm$ |

The stiffness of the thin filaments $k_{thin}$ were originally estimated in [36] from 1 $\mu$m long segments of rabbit skeletal muscle to be 65 pN/nm via deflection of a micro-needle under a

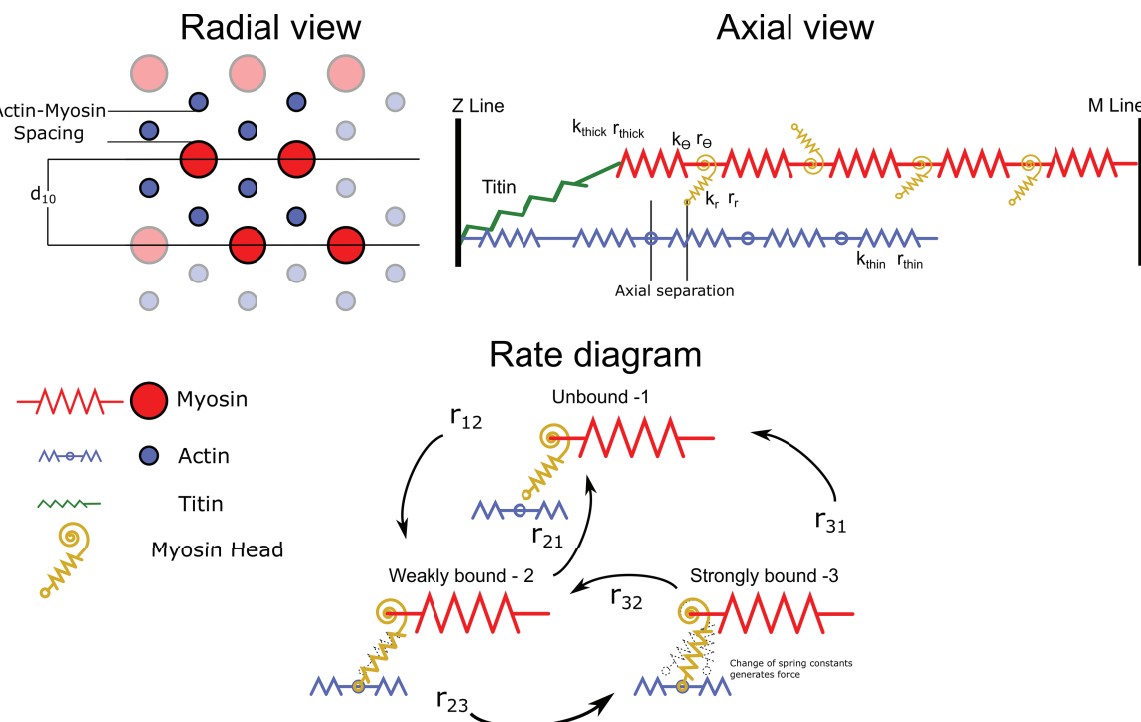

**Fig 2. Half sarcomere geometry and spring element stiffnesses.** The geometry of the spring lattice defines repeating motif that models the half sarcomere. Radial View: A cross-sectional view of the half sarcomere, showing the four thick filaments and 8 thin filaments present in model. The $d_{10}$ spacing is the lattice spacing of the crystal unit cell, measured by x-ray diffraction [13]. The actin-myosin spacing (minus the diameter of the thick and thin filaments) is the main parameter we vary in the model. Bold filaments indicate the 4 thick and 8 thin filaments present in model, while shaded filaments indicate connections made through periodic boundary conditions. Axial View: A 2-D longitudinal view of a segment of a thick filament and one thin filament with which it interacts. Each myosin head faces a certain actin-containing thin filament with which it can potentially bind. For clarity, we only show one thick and one thin filament, and only a few of the 720 crowns and actin binding sites. Titin attaches from the end of the thick filament to the z-disk where the thin filament also attaches. Rate Diagram: The thick and thin filaments are composed of series spring elements of stiffness $k_{thick}$ and $k_{thin}$ taken from empirical estimates. Equilibrium lengths are $r_{thick}$ and $r_{thin}$. Each myosin head is governed by a three-state kinetic model, but the free energy of each state is modified by the strain on the head. We use a two spring model for myosin composed of a torsional spring at the base ($k_{\theta}$ and $r_{\theta}$) and a linear spring in the arm ($k_r$ and $r_r$), as in [12]. The power stroke is mechanically represented by a change in the rest angle $r_{\theta}$ and length $r_r$ of the myosin motor.

microscope. The stiffness of the thick filaments comes from the observation that thick filaments are about 150% stiffer than thin filaments, as seen by strain changes in the thick and thin filaments via x-ray diffraction of frog skeletal muscle [37]. The repeat distances of 38.7 and 43 nm are then used to scale the stiffness of each segment of the two filaments [24].

Myosin binding during muscle contraction has been modeled with many different numbers of states [38–40], but based on prior models and because we primarily wanted to look at the effect of myofilament lattice structure on the force production step we focused on a 3-state model where myosin heads can be: 1-unbound, 2-weakly bound, and 3-strongly bound. Crossbridges are modeled by a torsional and linear spring, and conformational changes in the crossbridge cycle are represented mechanically as a change in the equilibrium angle and equilibrium length of the torsional and linear springs which comprise the myosin motor [12]. The weakly and strongly bound equilibrium locations of the myosin head come from electron tomography of quick frozen muscle of insect flight muscle [41,42].

Due to the 2 spring system of the myosin head, the radial direction becomes important when considering crossbridges force and binding probability.

$$U_{W,S} = \frac{1}{2}k_r(r - r_{W,S})^2 + \frac{1}{2}k_\theta(\theta - \theta_{W,S})^2$$
$$F_{W,S} = k_r(r - r_{W,S}) + k_\theta(\theta - \theta_{W,S})$$

$k_r = 5 KT nm^{-2}$
$k_\theta = 40\ KT rad^{-2}$
$r_W = 19.93\ nm$
$r_S = 16.4\ nm$
$\theta_W = 47.16\ rad$
$\theta_S = 73.2\ rad$

Here, we show the potential energy (U) and force (F) for a myosin head, with k and r indicating the stiffness and set length, and the subscripts $\theta$ and r representing the torsional and linear springs, respectively. The subscripts W and S represent the weak and strong states (equivalently, states 1 and 2).

This model also incorporates titin, a protein filament which attaches the thick filaments to the Z-disk, which defines the end of the sarcomere [32]. Each titin filament is connected to each of the four myosin-containing thick filaments at one end, and to the z-disk at the location where the actin-containing thick filament intersects the z-disk. Each titin filament therefore exerts a radial and axial force on the lattice. The force of titin is given by the equation $F_{titin} = a \cdot e^{b \cdot \Delta L}$, as in other models [2,3,32]. For the parameters $a$ and $b$, we used the same parameters as in [32]. In real muscle the stiffness of titin is thought to change with $Ca^{2+}$, and is increasingly recognized as an important contributor to muscle function [43], and it has been suggested that titin stiffness could significantly affect work [32]. Although titin is present in the model, in the current implementation does not include activation-dependent changes. Furthermore, titin is not present invertebrates like *M. sexta*, although a number of proteins such as sallimus, kettin, and projectin have been identified and serve an analogous function [44,45].

$$F_{Titin} = ae^{b\Delta L}$$

$a = 220\ pN$
$b = 0.0045\ nm^{-1}$

At the beginning of each time step, transition probabilities are calculated for crossbridge binding and state transitions based on the current state of each myosin head and its distance to the nearest thin filament binding site. The axial force on each node is calculated as the axial force from attached crossbridges as well as the axial force from displaced neighboring nodes. To solve for the equilibrium state of the half sarcomere, each node's axial location is iteratively adjusted so that the instantaneous axial force on each node is zero. The net axial force is then calculated as the axial force exerted by the node nearest the m-line on each thick filament. In real muscle, the radial forces are expected to do the same [8–10], in principal requiring a similar radial force balance. Although in our model there is no radial restoring force, we can explicitly prescribe the lattice spacing based on experimental data from x-ray diffraction to try and account for it.

## 2.3. Actin-Myosin spacing and $d_{10}$

The purpose of this study was to see if changes in the actin-myosin spacing in a half sarcomere model could modify work output in the spatially explicit model. In real muscle, the resting

and activated lattice spacing arises from the net force balance within the myofilament lattice which in turn affects myosin binding probability. However, this causal loop cannot be captured in current models because the forces of anchoring proteins at the z-disks are not included. So we prescribe lattice spacing based both on experimental measurements and systematic parameters sweeps and measure its effect on macroscopic force. Here we describe the relationship between the measured $d_{10}$ lattice spacing and the model parameter varied in the study.

Since the actin-myosin arrangement in muscle is highly ordered, x-ray diffraction can be used to measure the $d_{10}$ spacing. However, $d_{10}$ is a measurement of the size of the crystallographic unit cell, not a direct measurement of the actin-myosin spacing. It is however, proportional to the actin-myosin spacing, with the proportionality constant depending on the type of muscle. Vertebrate muscle, invertebrate limb muscle, and invertebrate flight muscle all have different proportions and arrangements of actin relative to myosin [7,9,20]. In vertebrate muscle, the actin-myosin spacing is given by $\frac{2}{3}d_{10}$ and in invertebrate flight muscle it is $\frac{1}{\sqrt{3}}d_{10}$. Vertebrate $d_{10}$ spacings are typically in the range from 35-40 nm, whereas invertebrate $d_{10}$ spacings tend to be larger, ranging between 40-50 nm. In order to facilitate the replication of the work presented here, we report the results of our simulations in terms of the actin-myosin spacing (after subtracting the radius of the thick and thin filaments which are 8 nm and 4.5 nm, respectively), which is the actual argument used in the model, and not the $d_{10}$, which is the physical measurement (Fig 2, Radial View). Therefore, in this model, the actin-myosin spacing is given by $AM_{spacing} = \frac{1}{\sqrt{3}} \cdot d_{10} - 8 - 4.5$. We centered our simulations on an actin-myosin spacing of 15 nm, which corresponds to a $d_{10}$ of approximately 47.5 nm in invertebrate muscle, the average value for *M. sexta* [30], and a $d_{10}$ of 41.2 nm in vertebrate muscle. Therefore the actin-myosin spacings we examine cover a large range of physiological relevant ranges of $d_{10}$ for both vertebrate and invertebrate muscle.

## 2.4. Rate functions

Rate equations for earlier versions of these spatial explicit models were originally established by fitting force under constant velocity data in [46] to a model in which crossbridges were represented by linear (axial only) springs. These rates were subsequently adapted in [24,25] to include dependence on crossbridge stiffness, and again in [11,12] to incorporate the radial component of the myosin heads.

The origin of the large negative work in the previous models (Fig 1) arises from many crossbridges being strained in unphysiological conditions. During a single work cycle at physiological strain velocities, a large population of crossbridges in the prior models transition to the loosely bound state $s_2$ even when strained at 20 nm, far from their equilibrium strain. They remain attached for some time, being further strained to ≈45 nm. This is substantially larger extensions than what a crossbridge should experience, which should be less than 10 nm during rapid shortening [39,46]. These abnormally strained crossbridges generate large amounts of negative (lengthening) force during shortening. These loosely bound crossbridges are not binding from an unbound state ($s_1$) but rather are reverting from the strongly bound state ($s_3$). This is because the $r_{31}$ rate does not increase rapidly enough at high strains, and reverse power stroke rate $r_{32}$, increases around –20 nm. While this regime of extreme, unphysiological strains were unlikely to have been explored in previous simulations of the model that consider isometric conditions, they prevent realistic force under dynamic conditions.

The inappropriate reverse transition to $s_2$ and persistence in that state comes from the model exploring the tails of the rate functions. In particular, the unbinding rate $r_{21}$ is the ratio

of the binding rate $r_{12}$ and the difference in free energies between two states of the expression $\exp^{U_1-U_2}$ (Fig 3). The falloff of $r_{21}$ is too slow relative to $\exp^{U_1-U2}$, which causes the unbinding rate $r_{21}$ to be 0 at extreme strains, when it should be rapidly rising. This meant that when tightly bound crossbridges revert from the strongly bound to the loosely bound state, instead of nearly instantly dissociating, they instead became negatively strained up to 40 nm during shortening. Similarly, loosely bound crossbridges would become positively strained during lengthening. The large forces caused by these highly strained crossbridges opposing length change in the sarcomere was the major cause of the negative work being done.

Because we wanted to maintain consistency with the previous instances of the spatially explicit model as much as possible, we sought to change the behavior of the rate functions by making rates steeper at higher strains without substantially changing their behavior at low strains. Comparing to the rate equations which were originally fit in [39,46], we saw that the binding rate $r_{12}$ exponentially decreases with increasing distance from the binding site just as in later versions of the model. However [46] also added a baseline rate of .005 ms$^{-1}$ to $r_{12}$ which is not present in the earlier versions of the model. At first glance this seems nonphysical, since it implies that crossbridges have a chance to bind at any axial distance. However the magnitude is too small to practically change $r_{12}$ significantly, and when we re-examine the $r_{21}$ rate, this baseline offset in $r_{12}$ corrects the problem with the binding rates exponential falloff, which enforces an infinite well in the $r_{21}$ rate without substantially changing binding rates in the working range of the myosin head. The transition rates used here, based on those in [12] are given by the following equations:

$$r_{12} = \tau * e^{-d^2} + .005$$
$$r_{21} = \frac{r_{12}}{\exp\left(U_0 - U_1\right)}$$
$$r_{23} = A * \left(1 + tanh(C + D(U_1 - U_2))\right)$$
$$r_{32} = \frac{r_{32}}{\exp\left(U_1 - U_2\right)}$$
$$r_{31} = G\sqrt{U_2} + H$$
$$r_{13} = 0$$

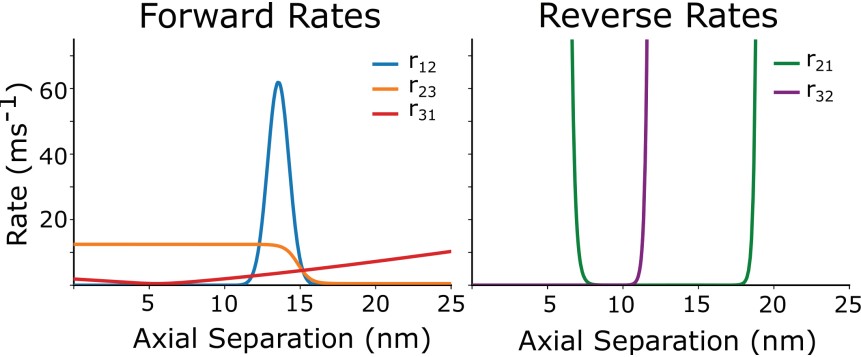

**Fig 3. State transition rates as a function of axial separation.** Axial separation is the axial distance between the origin of a myosin head and the nearest actin binding site. Left) The forward rates $r_{12}$, $r_{23}$ and $r_{31}$ rate. Right) Here we show the reverse rates $r_{21}$ and $r_{32}$. The rate $r_{13}$ is defined to be 0.

Here, $U_i$ is the free energy in the $i^{th}$ state, $d$ is the distance from the myosin head to actin binding site, and the rate constants $\tau$, $A$, $C$, $D$, $G$, and $H$ are chosen so that the function has units of $1\ ms^{-1}$, and the functions yield transitions consistent with previous models [12,24, 25,31] and experimental data [46]. Probability of a transition is calculated from the rate as $1 - e^{-r_{ij} \cdot dt}$, where $dt$ is the time step in the simulation. In our simulations, $\tau$=72, $A$=.8, $C$=6, $D$=.2, $G$=.6, and $H$=.02. Reverse rates are defined by the equilibrium equation $r_{ji} = \frac{r_{ij}}{e^{U_i - U_j}}$.

While this change was able to account for much of the negative work being done in work loops simulations, we still found that the $r_{21}$ was not tightly constrained compared to previous incarnations of the model [25,46], causing crossbridges to become nonphysically strained. While individual rate functions could be adjusted, the overall pattern is that myosin heads tend to remain in either $s_2$ or $s_3$ at unreasonably large strains. This is consistent with an underestimation of the effective stiffness of the myosin head. We therefore stiffened the myosin head's torsional spring by a factor of 10 compared to the previous model. This affects the $r_{21}$ rate since it is dependent on the free energy of the myosin head, which is dependent on the stiffness of both spring elements, and also makes the $r_{31}$ rate steeper [12,24,25]. We chose to increase the torsional spring stiffness since it is the dominant contributor to the steepness of the rate equations in the axial direction.

After these changes we found that the model produced much less tetanic force than the peak tetanus force of *M. sexta* DLM. We also found that the dominant contributor of force was from the loosely bound state, while the tightly bound state contributed little net force. To more closely match physiological data, which suggests the average steady-state force of a crossbridge should be about 8-10 pN under isometric tetanus [47], we increased the stiffness of the myosin head's linear spring by a factor of 4, and the power stroke rate constant by a factor of 10. Besides more closely matching the average force of a crossbridge, increased binding might be expected to match to data from invertebrate flight muscle because the original model in [46] was derived from rabbit psoas, a slower muscle than *M. sexta* DLM flight muscle [48,49]. Although the stiffness we use is larger than what has been reported from single molecule experiments, these experiments have been suggested to underestimate stiffness compared to the *in vivo* case [24,47,50].

## 2.5. Actin permissiveness parameterizes $Ca^{2+}$ and tropomyosin dynamics

In passive real muscle, actin bindings sites are obscured by tropomyosin, which wraps helically around actin and is regulated by the troponin complex of proteins. When a muscle is activated, $Ca^{2+}$ rapidly floods the sarcomere, binds to troponin C, which causes a conformational change in tropomyosin, allowing myosin heads to attach. When $Ca^{2+}$ is pumped out of the contractile lattice, tropomyosin reverts to its original confirmation, preventing myosin binding and force generation. In the model presented here, this entire process is parameterized in the model by a single 'actin permissiveness' value which is bounded from 0 to 1 and represents the availability of an actin binding site for potential myosin binding. A value of 0 indicates $Ca^{2+}$ concentration is too low to cause any of the actin binding sites to be unblocked, and therefore the sarcomere is totally passive, while 1 indicates $Ca^{2+}$ concentration is high enough that all binding sites are accessible, resulting in tetanic behavior. The actin permissiveness is the same for each binding site in the sarcomere even though the binding probability of a given site will depend both on this and the spatial arrangement of available myosin heads. The actin permissiveness we used in active workloops and twitch follows the equation $AP(t) = AP_{max} \cdot e^{((t^a - t_p)/w)^2}$ with $a = 0.73$, $t_p = 40 ms$, and $w = 14 ms$ (Fig 4).

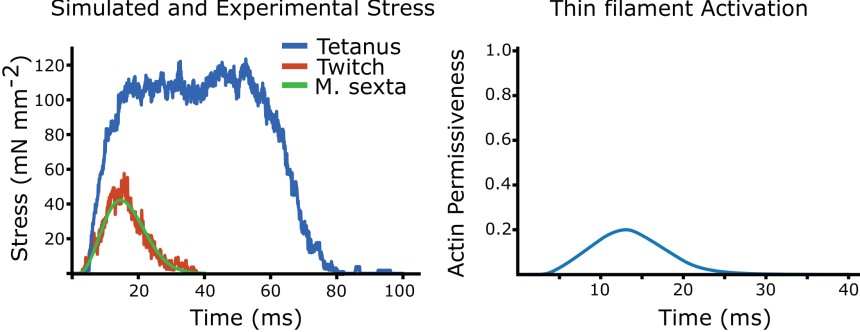

**Fig 4. Simulated twitch and tetanus.** On the left we plot the isometric tetanic force, simulated twitch force, and experimental twitch force from *M. sexta* dorsolongitudinal muscle (DLM). In tetanic simulations, the activation level is set to 1. Right shows the activation profile used to simulate the twitch force. This activation curve is also used in all the following work loop simulations except passive, for which the activation is set to 0.

### 2.6. Activation profile was found by matching to twitch force

Since work loops are cyclically activated, we needed to define a periodic function for the actin permissiveness, or activation curve, for the sarcomere. We set the shape the actin permissiveness curve as two exponential functions representing influx and re-uptake of $Ca^{2+}$. We then simulated an isometric twitch by choosing the influx time and half life of $Ca^{2+}$ re-uptake such that the rise, fall, and peak force during model response matched the twitch data we recorded from *M. sexta*. We recorded the twitch force by mounting the thorax between a dual-mode muscle lever (model 305C, Aurora Scientific, Aurora, Canada) and a rigid block and severing all muscles except the down-stroke muscles, similar to [27]. The simulated tetanic force and twitch force are shown in figure 4, as well as a twitch we recorded from *M. sexta* DLM. We used this same activation curve in all following work loop simulations.

### 2.7. Work loop protocol for phase sweep

After tuning our model to the twitch and tetanus data we recorded from *M. sexta*, we wanted to test if it could capture realistic levels of mechanical work under dynamic, physiological conditions. We simulated these work loops with a peak-to-peak strain amplitude of 10% and at a frequency of 25 Hz and varied the phase of activation, which is the moment of activation relative to the strain cycle, similar to what was done experimentally in *M. sexta* in [27]. Changing the phase of activation causes net work to smoothly transition from positive to negative depending on when during the strain cycle myosin heads are actively recruited, with a phase of 0 corresponding to the start of shortening. Each trial included 20 periods, and work was calculated for each period and averaged to obtain means and standard deviations. We initially kept a constant lattice spacing of 15 nm, which would correspond to a $d_{10}$ spacing of 47.5 nm in *M. sexta*.

## 3. Results

### 3.1. Simulated work-phase sweep captures main features of *M. sexta* work-phase relationship

Whereas prior models that incorporate explicit radial strain dependence did not generate any net positive work and were multiple orders of magnitude away from predicting force under dynamic conditions (Fig 1), the revised spatially explicit model produced a strong match to

physiological work loops at all phases. At a phase of activation of 0 - which we define as the start of shortening and the DLM's downstroke, which is also the average *in vivo* phase for hovering in *M. sexta* - our model produced $0.6 \pm .2$ J kg$^{-1}$ (mean $\pm$ s.d.), compared to $1.6 \pm .27$ J kg$^{-1}$ in *M. sexta* whereas the unaltered model predicted -230 J kg$^{-1}$. At a phase (0.8) that maximized positive mechanical, our updated spatially explicit model produced $1.06 \pm .28$ J kg$^{-1}$ compared to $2.93 \pm .59$ J kg$^{-1}$ *in vivo*. During phases of activation around the transition from the end of shortening to the beginning of lengthening (0.5), the model produced more negative work than *M. sexta*. For example, the model produced $-3.5 \pm 0.5$ J kg$^{-1}$, compared to $-1.9 \pm .4$ J kg$^{-1}$ *in vivo* (mean $\pm$ s.d.). Despite not being explicitly tuned to match the dynamic conditions of work loops, the model both captures work output to within a factor of 3 (compared to a factor of >100) and shows a phase dependency that matches *in vivo* expectations.

Comparing the simulated work loops with real work loops from *M. sexta* [27], there are several notable differences. First of all, there is a large passive component of force in real muscle which is not present in the model. This can be seen from the ramp in force as muscle strain increases, with the passive stress being about 20 mN $mm^2$ larger at maximum strain (Fig 5). Because the passive component of force in real muscle is much higher than in our model, we show also *M. sexta* work loops which have had the passive component of force subtracted (5). We found at the *in vivo* phase of activation for *M. sexta* of 0 (the start of shortening), that peak passive-subtracted force occurred 5 ms after activation occurred, whereas in simulated work loops the force rose much slower, only peaking 20 ms after activation. At a phase of activation of 0.4 (just before the transition from shortening to lengthening), the force in the simulated work loops rises much faster and higher during the first few milliseconds than in the passive subtracted, however they both exhibit the same plateau of force during lengthening. At a phase of 0.8, the force in *M. sexta* work loops is considerably higher than that of simulated work loops, with *M. sexta* work loops producing 100 mN/mm$^2$ compared to a peak force of only 40 mN/mm$^2$ in simulated work loops (Fig 5).

Many of these differences likely arise from not specifically matching the model to replicate *M. sexta* parameters. One possible avenue of for future research would be to examine if species-specific structural differences could give tighter fits to specific datasets. For example, we should expect variation in the actin:myosin ratio, the orientation of the repeating lattice unit, and the presence of other active filaments and regulatory proteins influence force production under dynamic conditions. Notably, the passive stiffness of titin has been shown to influence the amount of crossbridge binding and force in a spatially explicit muscle model under isometric contractions at high strain [32]. Since the passive component of our model is so low, increasing the stiffness of titin a significant amount could have a large impact on mechanical work. While elaborations could be made to make the updated spatially explicit model more like other specific systems, the fundamental formulation here is sufficient to test if structural variation can drive large changes in work output under physiological conditions. Our goal was not to optimally reconstruct work done by a specific insect muscle in a specific context, but rather to obtain a model that has reasonable behavior of insect skeletal muscle under dynamic, oscillatory conditions and then interrogate if lattice spacing can modulate this work in a significant way.

### 3.2. 1 nm spacing changes can generate positive or negative net work

The updated model allows us to now test if small differences in the radial spacing between filaments can modulate muscle mechanical work, as suggested in [20]. After getting a reasonable phase sweep at 15 nm, we simulated work loops at 14 nm. In invertebrate flight muscle, this

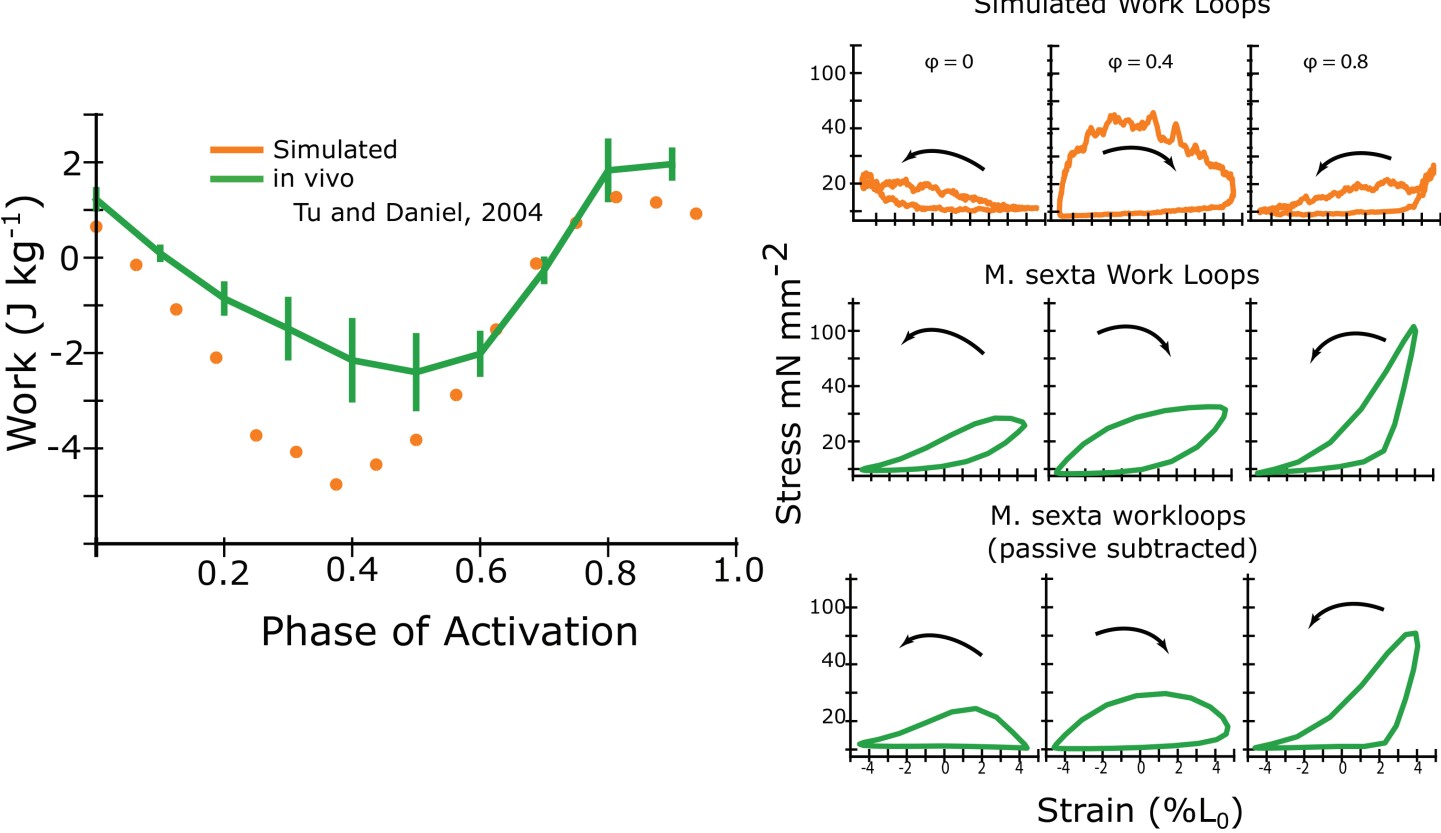

**Fig 5. Simulated net work vs. phase of activation.** Left) We plot the net work vs. phase of activation produced by the updated model (orange) as well as the measured *in vivo* net work for *M. sexta* (green), first published in [27]. The phase of activation is the time of activation relative to the length cycle, with $\phi = 0$ being defined as start of shortening. Right) We show example simulated work loops at phases of activation of 0, 0.4, and 0.8 (orange). We also show real work loops from *M. sexta* at the same phases for comparison (green). Because the passive component of force in real muscle is much higher than in our model, we show also *M. sexta* work loops which have had the passive component of force subtracted. Passive work loop data was collected new for this work, since it was not available from [27], using a very similar experimental rig and protocol.

would correspond to a $d_{10}$ change of 47.6 to 45.9, a 1.7 nm difference. We found that under these conditions, at a radial filament spacing of 14 nm the net work was negative (-0.74 ± 0.14 J kg$^{-1}$), while at 15 nm, the model produced net positive work (0.72 ± 0.14 J kg$^{-1}$) (Fig 6). A single nanometer difference in radial filament spacing can cause a switch in the sign of the model's work output.

We next extended the simulation to radial filament spacings from 12 to 17.5 nm, again keeping radial filament spacing constant throughout the entire work loop. At the *in vivo* phase of activation (start of shortening), the lattice spacing had a net work peak at 16 nm (Fig 7, $\phi = 0.0$, red). As lattice spacing increased from 12 to 16 nm, net work changed from -4.2 J kg$^{-1}$ to 1.3 J kg$^{-1}$, increasing positive work by 1.3 J kg$^{-1}$ nm$^{-1}$. Similarly, at a phase of activation of 0.85, the net work peaked at a radial filament spacing of 15.75 nm, with net work increasing 3.0 J kg$^{-1}$ nm$^{-1}$ from 12 to 16 nm. In contrast at a phase of activation of 0.15, net work only slowly increases with radial filament spacing, and never peaks over the range we examined. The peak in the phase of activation occurs at a radial filament spacing equivalent to a $d_{10}$ of 49 nm, while the recorded mean $d_{10}$ spacing in *M. sexta* is 47 nm [30].

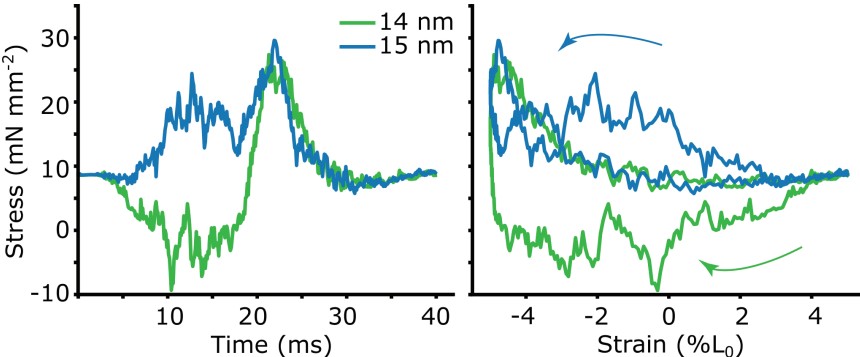

**Fig 6. Brake-like (net-negative) and motor-like (net-positive) work loops with 1 nm lattice spacing change.** We show stress vs time and stress vs strain (work loop) simulated at a constant actin-myosin spacing of 14 *nm* (green) and 15 *nm* (blue). Each trace consists of 20 work loop which have been averaged together.

### 3.3. Isovolumetric and *in vivo* lattice spacing dynamics increased net work by 10–20%

We next wanted to show how the net work would depend not only on the mean offset of the lattice spacing, but on the amplitude of the spacing change. In many muscles the lattice spacing in not constant, but depends on the length of the sarcomere [13,14,30]. We wanted to test how work would be influenced when we made the actin-myosin spacing depend on sarcomere length. To start with we chose to make the lattice spacing isovolumetric with length change. We then prescribed a time course of lattice spacing change similar to what has been recorded in *M. sexta* by x-ray diffraction during work loops [30]. We then compared the net work under these different conditions- constant lattice spacing (termed isolattice), constant volume (isovolumetric), and *in vivo* lattice spacing changes. Fig 7 shows the time course of lattice spacing changes for the different conditions, where $LS_0$ is the lattice spacing at the mean strain.

We simulated work loops at 25 Hz with the same activation and strain pattern used in work loops as above, and 10% peak-to-peak strain amplitude. Each point in figure 7 is the average of 20 periods of cyclical activation and strain. Isovolumetric conditions indicate the lattice spacing changed with length according to the equation $\Delta d = d\left(1 - \left(1 + \frac{\Delta L}{L}\right)^{-\nu}\right)$, where $d$ is the $d_{10}$ spacing, which we then convert to face-to face actin-myosin spacing, and $L$ is the length of the simulated half sarcomere. The Poisson ratio is given by $\nu$, and $\nu = .5$ indicates isovolumetric changes. The *in vivo* lattice spacing changes have a time-varying Poisson ratio. We simulated here three phases of activation, 0, 0.8 and 0.2, where we define a phase of 0 as the start of shortening. For comparison, in *M. sexta*, the average *in vivo* phase during hovering is 0 , with $\phi=0.85$ and $\phi=0.15$ being the approximate limits of the *in vivo* range during flight [51].

We found that while the mean spacing was a much more dominant factor in determining net work overall, the time course of the lattice spacing change could still have a small affect on net work. For example at a phase of activation of 0.2, activation begins at 8 ms after the start of shortening and peak actin permissiveness occurs 13 ms later (at 21 ms). Maximal activation therefore coincides with the peak lattice spacing change in the isovolumetric and *in vivo* cases. This increases net work by a small amount. In the case of isovolumetric change the work enhancement over constant lattice conditions is approximately constant between

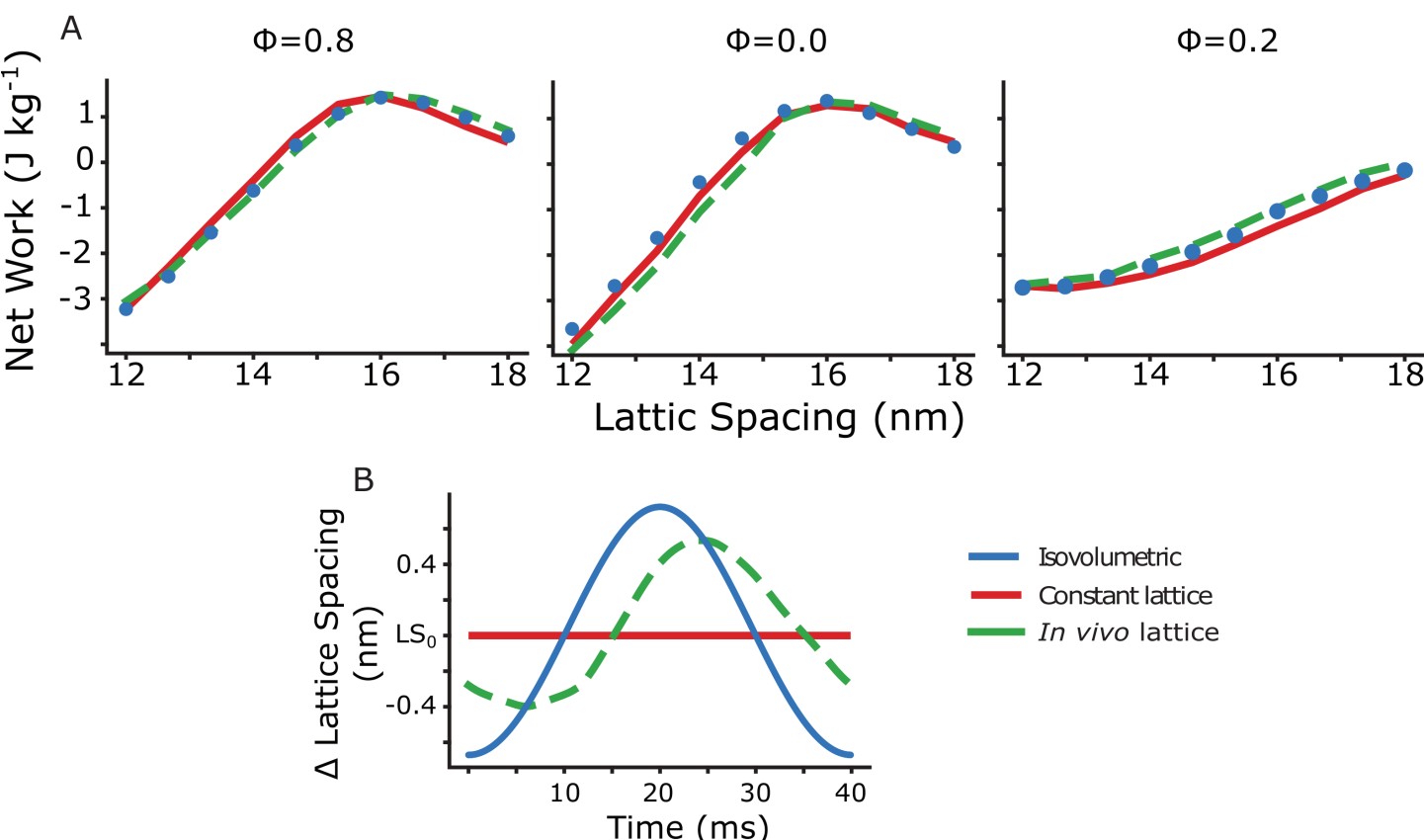

**Fig 7. Net work vs. lattice spacing under isovolumetric lattice changes, constant lattice, and *in vivo* lattice changes from *M. sexta*.** Top) We show the net work at phases of activation of 0.8, 0.0, and 0.2 under conditions in which the lattice spacing was either constant (isolattice, red), changed with sarcomere strain with a Poisson's Ratio ratio of 0.5 (isovolumetric, blue), or was prescribed according to the *in vivo* lattice spacing changes found from *M. sexta* (*in vivo*, green). A phase of activation of 0 is the average *in vivo* phase of activation during hovering and is the start of the downstroke of the wings, and the start of shortening of the muscle. Bottom) shows the prescribed lattice spacing changes in the different cases centered on $L_0$, with the *in vivo* lattice spacing changes derived from [30].

lattice spacings of 14 to 18 nm and increases by an average of 0.22 J kg$^{-1}$. Under *in vivo* lattice spacing changes, the net work enhancement is larger (average of 0.35 J kg$^{-1}$ over 14-18 nm), even though the peak lattice spacing change is smaller, possibly because the lattice spacing change is phase advanced compared to isovolumetric changes. This would allow for a larger mean lattice spacing for the portion of the work loop following peak activation. At a lattice spacing of 15 nm, this represents an increase in positive work of 12% for isovolumetric and 21% for *in vivo* lattice spacing changes compared to constant lattice. In contrast, the net work changes during a phase of activation of 0.8 are minimal for the three cases, since peak activation would occur at around 5 ms, when differences in lattice spacing between the three conditions changes are smaller. Had the phase shift between the *in vivo* lattice spacing changes been larger, we might have seen larger dependencies on time course of lattice spacing change suggesting that lattice spacing dynamics may have larger effects during large strain and high-frequency behaviors.

### 3.4. Cross bridge stiffness can attenuate or accentuate net work dependence on lattice spacing

Because we expect the stiffness of the springs composing the myosin heads to affect net work, we wanted to see how the stiffness of the linear and torsional springs affected the amount of work generated by the half sarcomere simulation (Fig 8). First, we calculated the net work for the default state of the model ($k_r$=16 pn · nm$^{-1}$, $k_\theta$=4000 pn · nm · rad$^{-1}$), and found the region of greatest net work was at lattice spacings between 15 and 16 nm and phases of activation between 0.8 and 0.1 (end of lengthening to mid-way through shortening) and the region of minimum net work was between lattice spacings of 12-15 nm, and phases of activation of 0.3 to 0.5 (midway through shortening to start of lengthening).

We next simulated the same range of lattice spacings and phases of activation, but set the stiffness of the linear and torsional components to ±50% the default values (Fig 8). We found that changing the linear stiffness affected the net work most for phases of activation near the start of lengthening ($\phi$=0.5), with increasing stiffness being associated with lower net work. The torsional stiffness had the most effect at lower lattice spacings. Stiffening the torsional spring led to a further decrease in network at phases of activation near the start of shortening ($\phi$=0), but greater net work at the start of lengthening.

### 3.5. Titin exponential stiffness changes did not affect net work

We also chose to examine how the stiffness of titin might affect net work. In the model, the elastic coupling between any elements may affect the compliant realignment of myosin heads during force production [24]. It was shown previously that the stiffness of titin could affect the isometric force, since the realignment of the crossbridges could be more or less depending of the relative stiffness of titin [32]. Also, physiologically, titin and titin-analogs in muscle are thought to regulate lattice spacing dynamics [52,53]. Therefore, we also simulated half sarcomere work loops under varying exponential stiffness. The force of titin is here modeled as $F_{titin} = a \cdot e^{b \cdot \Delta L}$, and we set $a = 260$ pN and we varied the parameter $b$ from 4 - 10 $\mu m^{-1}$, as in [32], which covers the reported range of estimates for single titin molecules [54]. It was previously seen in [32] that isometric force was diminished when the half sarcomere was at lengths greater than 2.7 $\mu$m. In contrast, we did not find that increasing the stiffness of titin had a large impact on the net work 9, likely because we did not investigate the same sarcomere length range. Since we based our work loop simulations on *M. sexta* DLM, we chose a 10% peak-to-peak amplitude around a sarcomere length of 2.5 $\mu$m, which meant we did not examine the regime with sarcomere lengths large enough to cause reduced force in the force length curve [32]. Also, as indicated in Fig 5, the passive force present in our simulated work loops is much lower than in *M. sexta*, so it may be that the force of titin in the model is too low to see an effect.

## 4. Discussion

The updated spatial explicit model can simulate realistic scales of mechanical work under dynamic conditions and supports the hypothesis that nanometer scale changes in the myofilament lattice can significantly effect the mechanical output of whole muscle. Previously, it was shown that lattice spacing differences on the order of 1 nm in two muscles in the cockroach *Blaberus discoidalis* were associated with their different mechanical functions [20]. However, it could not be definitively shown that the lattice spacing differences observed were responsible for, rather than just correlated with modulating work. The updated model results demonstrate that differences in mean lattice spacing alone, even at the scale of a single

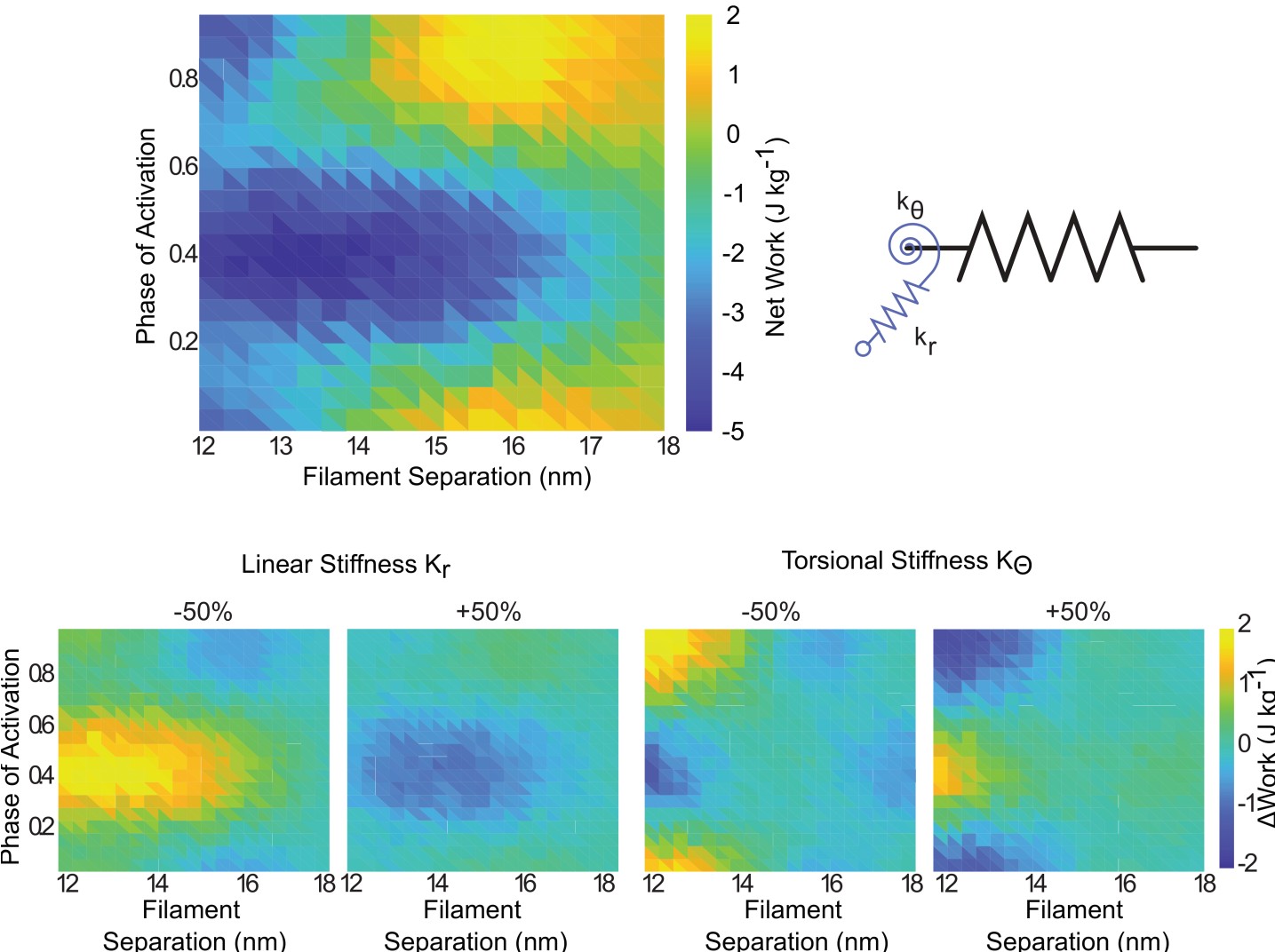

**Fig 8. Net work vs phase of activation and lattice spacing.** Top) We simulated work loops in the half sarcomere model at phases of activation of 0 to 0.95 in 0.05 increments, as well as over lattice spacings from 12 to 18 nm and plotted the new work for each condition. Bottom) We then simulated work loops over the same range, but with the stiffness of either the linear or torsional spring comprising the crossbridge head increased or decreased by 50% separately. Data shown in the bottom panel are shown as change relative to the top.

nanometer, can drive mechanical functional differences, for example switching a motor to a brake.

## 4.1. Lattice spacing and crossbridge stiffness mediate multiscale interactions that alter whole muscle function

Even though the amplitude of lattice spacing change over the course of one contraction cycle is only a few nanometers, it can have a large effect on force production because it affects the binding rates of all of the billions of myosin motors. Lattice spacing can also effect the force produced during the power stroke because of the amount of strain on the crossbridge and deformation of the filament backbone. These effects are sufficient such that changing lattice spacing alone can alter the emergent net mechanical work of a muscle and even change

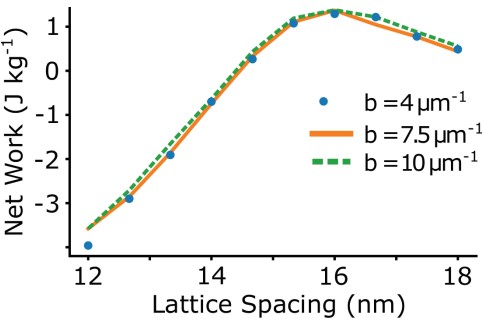

**Fig 9. Net work with different exponential stiffnesses of titin.** We simulated work loops in the half sarcomere model at constant lattice spacings of 12 to 18 *nm* with three different values for the exponential stiffness of titin. We found that the titin stiffness did not effect the net work under the conditions we examined.

its sign (Fig 7). The effect may even differ across sarcomere because the lattice spacing of a sarcomere will depend on the length of each sarcomere, which may not be uniform in the whole muscle [2,55]. This means that as muscle oscillates, the lattice spacing is an important determinant of muscle force.

In our model we prescribe lattice spacing and measure force. However, the lattice spacing in real muscle is due to the balance of the radial forces acting on the lattice and is not fully independent. These radial forces can arise from regulatory proteins such as dystrophin [56] or titin [57], as well as confinement forces from the z-disk, which are unmodeled in the current spatial explicit model. Fluid forces are also present in the lattice [58]. Crossbridges also contribute because of their radial forces. So there is a feedback between crossbridge binding and lattice spacing [8]. So lattice spacing and myosin binding interact, but we cannot simulate their full coupling without access to the other radial forces acting on the lattice. As with prior models [12], we therefore treat lattice spacing as an parameter or input to the model and binding and force production as an output. This is because we can experimentally measure lattice spacing, but when we sweep lattice spacing parameter space not all regions might be biologically accessible. As half sarcomere models become even more complete it would be interesting to allow both binding and lattice spacing to emerge.

In the case of the cockroach muscle there is a 1 nanometer lattice spacing difference at rest [20]. Presumably this difference arises from the different anchoring and radial force balance not due to myosin head binding. While we do not know how the lattice spacing's relationship with length is set in muscle, it seems to be muscle specific [20,26,30]. While we saw a large effect due to changes in lattice spacing offsets 5, we only saw a small change due to the time course of lattice spacing change 7, possibly since the amplitude of lattice spacing change is small. However, even though real muscle has a very complicated structure, including many more elements than are in our model, we are still able to show the potential for lattice spacing to affect net work.

A 1 nm lattice spacing change (1.8 nm $d_{10}$ change) in our simulations produces a 0.6 J $Kg^{-1}$ difference in work. This does not fully explain the $2.386 \pm 1.8$ J $Kg^{-1}$ difference in the two cockroach muscles [21], likely because we do not have all the model biophysical parameters available for cockroach muscle. We had to ground our model's behavior in twitch, tetanus, and work loop data taken from *M. sexta* [27] and parameter estimates from prior studies. *M. sexta* was also the only source of very detailed time-resolved measurements of the lattice spacing [30,58]. So the model is best considered as a generalized model of insect locomotor

muscle which we use to determine the scale of potential influence work production of particular factors. As more comparative measurements of skeletal muscle biophysical parameters become available we could use version of this model tuned to different organisms to directly compare drivers of different muscle functional differences.

## 4.2. 3D spatially-explicit models enable interpretation of muscle's multiscale effects

By changing the behavior of the binding rate kinetics, especially in relation to high frequency, large amplitude strain changes, we have significantly improved the spatially explicit models of muscle. We are now able to simulate work loops in a physiological regime with explicit radial dependencies. Generally, spatially explicit models of muscle allow for studying how the geometry and mechanical coupling of the myosin motors can impact force and work while incorporating interaction due to deformation of the myofilament lattice. Even when the myosin motors themselves remain unchanged, effective changes in their dynamics can occur due to multiscale interactions, for example, enhancing crossbridge binding by altering filament stiffness alone [24]. These kinds of models capture dynamics that mass action models alone are not able to account for these kinds of multiscale, emergent behaviors. While spatially explicit models can be more computationally intensive, machine learning methods can be used to develop emulators [59]. These emulators mimic the original model while being much faster and will catalyze broader use of these models in the future.

Myosin binding and lattice spacing also interplay with the stiffness of the filaments [12, 24,25,32]. In general, there is a trade-off in that high compliance in the thick and thin filaments allow more crossbridge binding, but less force per crossbridge [25]. Also, by increasing the stiffness of the myosin heads, thermal forcing in the unbound state is reduced, which can reduce the effective distance at which heads can bind. Higher stiffness, however, can increase the force that each crossbridge can produce. By altering the crossbridge stiffness in conjunction with the lattice spacing and phase of activation, we were able to test how crossbridge stiffness attenuated or accentuated the work landscape. Changes to stiffness had a much larger impact when the thick filament was closer to the thin filament, and the torsional spring mostly affected net work during active shortening, while the linear spring affected work during active lengthening (Fig 8). These kinds of interacting effects currently can only be shown in a spatially explicit model, where the geometry as well as the biophysics of present elements can be investigated.

Even with the refinements here, the spatially explicit model does not contain all possible factors contributing to muscle force. While sufficient to test general dependencies like the sensitivity of mechanical work at the macroscopic scale on nanometer scale lattice spacing, further refinements may enable these models to better match specific muscle conditions. For example, the effects of activatable titin could have a large effect on the amount of work produced. In [32] it was shown that by increasing the exponential stiffness of titin, crossbridge binding could be increased at high strains, however force of each crossbridge was lower. They also predicted that stiffening titin could also decrease the negative work produced. We were unable to see a difference in the net work produced in the model under the same stiffness values (Fig 9). However, titin is not simply a passive exponential spring, but may have $Ca^{2+}$ dependent properties [43,60]. By introducing activatable titin, for example by making the stiffness of titin dependent not only on length but also the actin permissiveness, we might expect an even more dramatic dependence of net work on lattice spacing.

However, even it is current form the model presents an opportunity to study how the geometry of other features of muscle structure affect muscle mechanical function. For example, it has been recognized that the thin-thick filament ratio and arrangement in different muscles can be very different. Vertebrate muscle has a 2:1 thin:thick filament ratio, invertebrate flight muscle has a 3:1 thin:thick filament ratio, and invertebrate limb muscle has a 6:1 thin:thick filament ratio [9]. Furthermore, crown packing can be different in various taxa. For example, the crown rotation angle can be 60deg, 60deg, 0deg, as found in [35] and simulated here, or 40deg [61]. While the specialization seen in various kinds of invertebrate muscle might be indicative of some functional consequence for these thin:thick filament packing patterns, it has not been investigated what this might be. Isolating the effect of different geometries in sarcomere structure would be very experimentally difficult, whereas in a spatially explicit model the geometry of crossbridge motors and actin binding sites can be examined.

## 5. Conclusion

We were able to show in a spatially explicit model with prescribed radial spacing differences that we could obtain physiological amounts of force and net work. We showed that the lattice spacing could affect the net work in such a model. This model provides a framework for examining how the biophysics and geometric arrangement of force producing crossbridges in muscle can scale through sarcomeres to the whole muscle scale.

## Acknowledgments

We would like to thank Tom Daniel, Dave Williams, Joe Powers, and Anthony Ascencio for their helpful discussions. We would also like to thank the University of Washington Center for Translational Muscle Research.

## Author contributions

**Conceptualization:** Travis Carver Tune, Simon Sponberg.

**Formal analysis:** Travis Carver Tune.

**Funding acquisition:** Travis Carver Tune, Simon Sponberg.

**Investigation:** Travis Carver Tune.

**Project administration:** Simon Sponberg.

**Software:** Travis Carver Tune.

**Supervision:** Simon Sponberg.

**Writing – original draft:** Travis Carver Tune.

**Writing – review & editing:** Travis Carver Tune, Simon Sponberg.

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
