## [Decision Letter · Decision Letter 0]

21 Jun 2024

Dear Dr. Tune,

Thank you very much for submitting your manuscript "Nanometer scale difference in myofilament lattice structure of muscle alters muscle function in a spatially explicit model" for consideration at PLOS Computational Biology.

As with all papers reviewed by the journal, your manuscript was reviewed by members of the editorial board and by several independent reviewers. In light of the reviews (below this email), we would like to invite the resubmission of a significantly-revised version that takes into account the reviewers' comments.

We cannot make any decision about publication until we have seen the revised manuscript and your response to the reviewers' comments. Your revised manuscript is also likely to be sent to reviewers for further evaluation.

Sincerely,

Daniel A Beard

Section Editor

PLOS Computational Biology

Daniel Beard

Section Editor

PLOS Computational Biology

Reviewer's Responses to Questions

**Comments to the Authors:**

Reviewer #1: Tune et. al. PLOS Review Comments

This paper describes a model of a sarcomere that was deployed to demonstrate how a small change in myofilament structure (geometry) can have a profound impact on function including work produced. The main message of the paper in possibly explaining previous experimental observations is of wide interest to the field, however, the presentation of the paper makes it hard to access. The following are specific comments:

Major comments

1. The introduction is a bit convoluted and longer than necessary. For example, the paragraph that starts on line 73 is already putting this manuscript into context, but then the paragraph on line 86 goes back into overall background. The introduction should end with a tight paragraph that clearly explains how the paper plans to address standing challenges in the field.

2. Figure 5 would greatly benefit if there was something else to compare it to. As it is, the plots could be interpreted as being far away from experimental data (none of them match) or if other models are not able to replicate the qualitative behavior, this could be interpreted as a significant step forward. As it is, the reader is left floundering.

3. The methods portion is lengthy - the language could probably be tightened for a more efficient section.

4. Overall, the presentation of the model and data is convoluted and difficult to follow due to typos, missing units, inconsistent terminology, lacking legends and captions, etc. Here are some examples and details:

a. The authors are inconsistent with their use of the terms “axial spacing,” “actin-myosin spacing,” “radial spacing,” and “lattice spacing.” Initially “axial” is used to describe strain, force, or overlap in the direction parallel to filaments and “radial” is used to define force or spacing between thick and thin filaments (Line 22-33; line 136-150; line 228). “Axial” is defined similarly with respect to “lattice spacing” later in lines 93-102 and Fig. 3. Figure 2 then defines the term “actin-myosin spacing” as it is mainly used later in the paper, much before it is discussed in the text (Section 2.6). At a minimum, Fig. 2 should be referenced in the discussion of actin-myosin spacing in the methods; Section 2.6 should be moved immediately after Section 2.2 to improve flow. The inconsistency lies at the start of Section 3.2, line 409 where the author refers to small differences in “axial spacing” of the myofilament lattice, where the term actin-myosin spacing should be used instead (referenced as lattice spacing later in paragraph). If the terms lattice spacing and actin-myosin spacing are indeed interchangeable, I recommend that only 1 be used for consistency throughout the paper (they are referenced separately in back-to-back sentences lines 424-425). This spacing is once again redefined as “filament separation” in Figure 8. The inconsistencies make the text very difficult to follow.

b. Several figures reference panels by letters, but the figures do not include the corresponding labels (Fig. 5). Other figures reference panels that are inconsistent with the panel labels (Fig. 2A and B). Yet other parts of the text reference panels that do not exist (line 280)

c. The lattice shown in Fig. 2A should include a legend. The states should be labeled by number in Fig. 2C as well as the transition rates between them so that it ties to the references made in the text.

d. More on typos and grammar in minor comments below

5. The authors claim that increasing torsional stiffness decreases network in areas already associated with negative work is inconsistent with the yellow/green region in Fig. 7C. This discrepancy should at least be mentioned in the discussion.

6. The discussion of titin’s effect on net work is minimal, and anticlimactic, especially given that the species this study is basing its simulations on does not actually possess titin. Further, the fact that myosin’s torsional and linear springs were increased by factors of 10 and 4 respectively from 2sXB to match M. sexta cross-bridge tetanic forces, and later in vivo passive stiffness is mentioned to be much higher than in the model and to have a significant effect on work loops (Fig. 5), it follows that a change in titin stiffness (and perturbations about this change) may demonstrate greater effects on net work as seen in Fig. 5. In fact, titin’s stiffness’s potential effect on work is directly mentioned in line 398-399. If titin’s stiffness is meant to be the concluding figure of this study, it warrants more investigation, otherwise, its discussion should occur earlier in the results section.

7. The first paragraph of the discussion sections (505-514) was one of the best parts of the paper that really brought home the major contribution of this work to the field. However, the rest of the discussion section was hard to follow and overly long (ex. page 19 paragraph 541-558 could really be condensed to a couple of sentences).

8. There is a lot of discussion on radial force in both the introduction and discussion, but it didn’t seem to be the main point of the paper.

Minor comments

1. M. Sexta experimental data is referenced in several captions, citations should be made available in the caption as opposed to having the reader return to the main text to find the references (3 mentioned initially in intro [26, 28, 29], only 1 mentioned elsewhere [28]).

2. How axial separation is calculated in Fig. 3 is unclear, the caption should include a clear definition. Additionally, the units for actin-myosin spacing are missing in the caption.

3. Why is Fig. 7A green dotted while a solid line in Fig. 7B. Mention that the in vivo lines are green in the caption to be consistent with the prior two conditions. L0 in the caption should be LS0 to match the axis in 7B.

4. Whether lattice spacing is constant or dynamic during simulations should be clearly mentioned in captions (e.g. Fig. 6 and Fig. 9)

a. Units in Fig. 6 caption

b. Fig. 9 caption: “The condition b = 4 um-1 is the condition used in all simulations above” appears to refer to the figure, not prior simulations. This standard titin stiffness should be mentioned in the methods where Ftitin is initially defined.

5. Phases mentioned in line 448 include significant figures not mentioned in Fig. 7A labels.

6. There are quite a few grammatical mistakes in the paper:

a. Lines 9-10 (page 2), sentence fragment – not clear what is meant.

7. There is also writing that is hard to follow as there are missing logical linguistic links to guide the reader. For example, the last sentence of the paragraph on line 32-33 (page 2) needs to be logically linked to the rest of the paragraph.

8. Missing citations on Line 323-325 and Line 376-377.

In summary, this work has the potential to be of interest to the field, but the manuscript would need to be significantly re-written prior to publication.

Reviewer #2: The authors demonstrate importance of spatially explicit muscle modeling and namely effect of distance in actin-myosin lattice. They adjusted the model from prior works by comparing the stress to a flying muscle of a moth. On the model, with work output profiles reminding those of model organism, the found, that first, any difference in (constant) lattice spacing (LS) greatly affects the muscle work output and second, the physiological time-varying lattice distance result in almost same net work as constant LS.

This seems to me it is more of a collateral effect than a driving principle. As such, it needs to be described, but its effect on simulations is probably low. Physiologically, if the lattice was observed to be different, then the spring lengths, stiffnesses and default angles would have also been different. Although this is acknowledged in discussion on line 528, this chicken-egg problem is substantial caveat of the paper and the imposed differences of the distance should be considered as hypothetical.

There might be an opposite use case for the model though - what is the optimal spacing given the chosen set of k and r? And, alternatively, what is the otpimal k and r, given the observed lattice spacing, which can be better measured? That said, although the developed model has a wide demonstrative potential and it is tempting to show all of its capabilities, I recommend the authors to focus on less.

Major comments:

1. An issue I was often facing throughout the paper is that the potential reader might not be familiar with the prior works, yet should still be able to understand the presented work. Namely, please explain in more detail the work loop at different activation phases and / or provide an explanatory figure, incl. sarcomere lengths. Is it that you shifted strain and activation? Can you explain why the maximal power output is at 0.8, while the physiological in vivo activation phase at 0 (line 419) is suboptimal?

2. Similarly, the modified model needs to be described in a little more detail, to have a basic understanding without the need of reading the prior works first. Namely the figure 2 can use some refinement to make it more illustrative.

a. Figure 2A: The figure suggests you are calculating interaction of a single myosin and 6 actins - is this correct? What role do the additional 2 actins and three myosins play then?

b. figure 2B - if the crowns are in triplets, the bottom- and top-facing heads should probably alternate down, top, down, top... correct?

c. Rates are not labeled in fig 2C - label the transition and rates accordingly

d. Please be explicit about the activation function - what parameters were tuned in the current work?

e. Please explain the role of k and r in generating the force. I assume it follows F~ (x-r)*k. But then the r of the thick and k of the thin filament have strange units and R of the thin filament is not defined (probably 0?)

f. Line 290: How does setting these parameters arbitrarily align with previous justification of the values?

g. Figure 6 caption: The stochasticity is not due to the model being spatially explicit, but the noise is high because the model is stochastic. Can the noise by tuned down with higher number of averaged periods? It is not common to see simulation results being more noisy than the measurement. Prior description of the model in the current paper does not make obvious whether the model is deterministic or not.

h. Line 346: please explain the relation of spacing to d10 - does it mean the radii of actin and/or myosin in vertebrate muscle is much smaller? Similarly line 412 - how the spacing of -1nm makes lattice -1.7nm?

3. Section 3.1: Consider moving the protocol description to the methods and concentrating on the true results here.

4. Further sections (3.4 and 3.5) are a bit confusing as of the point the authors are trying to make. Static parallel stiffness of titin should not affect the work in principle (or should it?). The conclusion on line 502 is not supported by these obsrvations - the model simulation simply did not reproduced this phenomenon, if it was present at all. Looking aside from the net work, do these experiments show difference in total force, contraction speeds or some other important factors of the muscle contraction?

Minor comments:

5. Line 117 - the 2sXB is not defined.

6. Model geometry - I do not understand why you chose such subset - there are only 6 active connections, as in 2A? Then the myosin heads to actin ratio should be 6:1 in 2B?

7. What is the poisson's ratio?

8. What is a protocol for fig 1? At which SL?

9. typo 231 and 262: 2xSB

10. line 295 - did you mean pN?

11. Line 339: typo piror

Reviewer #3: The manuscript by Tune and Sponberg extends a set of spatially-explicit models integrating radial and axial forces in the cross-bridge binding activity to explore the effects of muscle contraction during a simulated workloop. The authors compare radial compression of the myofilament lattice and stiffness values of the cross-bridge to show that changes in spacing and stiffness can augment or diminish the amount of work produced by the muscle. Authors compare these data to experimental data from Manduca Sexta, albeit somewhat qualitiatively. Clearly there are quantified numbers showing the differences between positive work production vs. negative work absorption, but it is difficult to compare the numbers directly. The main take away from the computationa simulations is the authors trying to show that lattice spacing is very important for dynamic contractility, work, and power production. This is an interesting idea.

Detailed comments are listed below:

Major:

Methods, section 2.1, page 4. It is a little unclear if the model being used or discussed solely accounts for axial force balance and force generation from time-step to time-step vs. axial and radial components of the calculations. From Fig. 2 it is clear there are axial and radial aspects of the cross-bridge properties, but it seems these axial and radial cross-bridge calculations do not translate into the axial force balances utilized to calculate the force produced during the simulation. Please clarify or advise.

Methods, Table 1. Characteristics or details related to the Thin Filament are not clear or may be missing proper units. Perhaps description of the table or a legend could be helpful.

Methods, page7, line 167: “Myosin head triplets are azimuthally distributed by 120° and adjacent crowns are rotated by 60°.” Please clarify or advise. Is this packing supposed to represent the insect flight muscles that are being simulated? I don’t know the insect flight muscle packing exactly, but this could be clarified or referenced. I may be incorrect, but this packing is not consistent with vertebrate cardiac, nor skeletal, muscle packing of the myosin crowns. Specifically, I think the degrees of rotation per crown appears incorrect.

Methods, page 7, line 176. In discussing or listing the stiffness values for the thick and thin filaments—these are all from vertebrate skeletal muscle measurements (I think as authors also reference). How might this compare to what is known about the insect muscle stiffness values the authors are trying to simulate or emulate with their workloops? For example, aren’t the stiffness characteristics of fruit fly and lethocerus muscles much stiffer than vertebrate muscles? Have the authors considered the implications of this for their simulations or the interpretation of the data in their study? Have the authors considered a set of simulations with much stiffer thick filaments and much floppier thick filaments, for example, to compare with the current simulations?

Methods, Page 11, Section 2.4 and Fig. 4B. The concept of actin permissiveness is very unclear. Is this some probability of binding for the whole system or for a single actin binding site etc. What gives the permissiveness a temporal dynamic that rises and then falls, or where is the mathematics driving tis behavior? How is this dealt with related to spatial components of activation vs. the probability of activation at any single binding site vs. activation of the system as a whole?

Results, Page 12, Section 3.1. Please remind or clarify the definition of the phase of activation timing. When it is related to which aspects of the wingbeat or the shortening and lengthening of the muscle. As in where is phase =0 and phase =0.5. I realize you define this related to Fig. 5 and the legend of Fig. 5, but something seems confusing about these assumptions or their implementations.

Results, pages 12-13. Is there any phase shift or viscous delay of the passive component of the workloop? Did the authors simulate the passive workloops to subtract them from the activated workloops? Since the model is purely elastic, likely not any temporal shifts or phase delays. Please comment on details or show the passive simulations if they are interesting and important to illustrate important aspects of these assumptions for subtracting the passive components .

Results: It is a neat idea and a nice set of simulations showing that lattice spacing changes can amplify or dissipate the net work throughout a dynamic contraction of a workloop. However, it is not clear why this occurs. Is it really as simple as the cross-bridge stiffness? How does this sensitivity to the cross-bridge stiffness scale or become affected by the kinetic rate parameters chosen to underly the cross-bridge binding and cycling? Have the authors considered a sensitivity analysis to further investigate these details for which portions of the chemomechanical cycle this finding stems from?

Minor:

Results, page 15, line 462. Missing units on the lattice spacing value(s).

Discussion, page 19, line 542. Work in this sentence seems to be missing units.

Discussion, page 21, line 619. Interesting ideas about the thick to thin filament ratios. There are also differences in the crown packing or myosin crown organization among taxa that would be helpful to comment or consider here too.

**Have the authors made all data and (if applicable) computational code underlying the findings in their manuscript fully available?**

Reviewer #1: Yes

Reviewer #2: Yes

Reviewer #3: Yes

PLOS authors have the option to publish the peer review history of their article (what does this mean?). If published, this will include your full peer review and any attached files.

Reviewer #1: No

Reviewer #2: No

Reviewer #3: No
---

## [Decision Letter · Decision Letter 1]

19 Nov 2024

PCOMPBIOL-D-24-00909R1Nanometer scale difference in myofilament lattice structure of muscle alters muscle function in a spatially explicit modelPLOS Computational Biology Dear Dr. Tune, Thank you for submitting your manuscript to PLOS Computational Biology. After careful consideration, we feel that it has merit but does not fully meet PLOS Computational Biology's publication criteria as it currently stands. Therefore, we invite you to submit a revised version of the manuscript that addresses the points raised during the review process. Please submit your revised manuscript within 30 days Jan 19 2025 11:59PM. If you will need more time than this to complete your revisions, please reply to this message or contact the journal office at ploscompbiol@plos.org. Please include the following items when submitting your revised manuscript:* A rebuttal letter that responds to each point raised by the editor and reviewer(s). You should upload this letter as a separate file labeled 'Response to Reviewers'. This file does not need to include responses to formatting updates and technical items listed in the 'Journal Requirements' section below.* A marked-up copy of your manuscript that highlights changes made to the original version. You should upload this as a separate file labeled 'Revised Manuscript with Track Changes'.* An unmarked version of your revised paper without tracked changes. You should upload this as a separate file labeled 'Manuscript'. If you would like to make changes to your financial disclosure, competing interests statement, or data availability statement, please make these updates within the submission form at the time of resubmission. Guidelines for resubmitting your figure files are available below the reviewer comments at the end of this letter. We look forward to receiving your revised manuscript. Kind regards, Daniel A BeardSection EditorPLOS Computational Biology Daniel BeardSection EditorPLOS Computational Biology

Feilim Mac Gabhann

Editor-in-Chief

PLOS Computational Biology

Jason Papin

Editor-in-Chief

PLOS Computational Biology

 **Journal Requirements:** 1) Please ensure that all Figure files have corresponding citations and legends within the manuscript. Currently, Figure 9 in your submission file inventory does not have an in-text citation. 2) Please upload your figures to the file inventory in a numerical order.

  **Reviewers' comments:** Reviewer's Responses to Questions

**Comments to the Authors:**

Reviewer #1: The authors have addressed most of my comments. A few minor issues remain:

1. Figure 1 - the way the arrows are set for the inset, it was very hard to figure out what was going on. Additionally the colors are very confusing where it seems that green means one thing in one panel and another in the other panel. The left panel should have a box that indicates what the inset is zooming onto.

2. Figure 2 - the legend inside the figure should also cover the symbols in the axial view and rate diagram.

Otherwise, I believe the paper will be of interest to the readers in this field, and the manuscript was much easier to read with the new revisions.

Reviewer #2: I want to congratulate the authors on substantially improving the paper. The aims are much clearer and the methods section are better understandable now. Still, I have found some minor issues.

The thick and thin filament parameters presented after line 163 are confusing. What is a and b here and why do the thick and thin filaments have different units? Are the labels flipped and k_thick and r_thick should be used instead?

Section 3.5 I do not see, how the passive properties are related to the muscle work output. The titin is modeled without hysteresis so it may not affect the work per se. Moreover, as admitted, the modeled passive force is minor compared to the one in vivo. Although this observation might be really interesting, it lacks any experimental support as well as mechanism in the model. I suggest dropping this whole section, noting in the end of section 3.4 that the change in simulated net work was not observed for a change in titin stiffness for the reasons mentioned above.

LIne 378: How did you subtract the passive force from fig 5? I have not found any explanation in the original work [27].

Regrettably, I was not able to run the code due to security limitations of my current computer. The repository seems to be more than 3 years without any update, please check the source code is up-to-date with the plots in the paper.

Fig 9 caption typo “three different vales”. LIne 300 The acronym DLM has been explained in a dropped paragraph. Please explain in text.

Reviewer #3: The paper is much improved.

With regard to the activation levels discussed in Fig. 4. Is the small activation level (i.e. maximal at 0.2 permissive value) in Fig. 4, right hand panel really actually used for all simulations? Thus, it is confusing about what happens with the tetanic stress response on Fig. 4 left side panel and what activation looks like related to that simulation response.

**Have the authors made all data and (if applicable) computational code underlying the findings in their manuscript fully available?**

Reviewer #1: Yes

Reviewer #2: Yes

Reviewer #3: Yes

PLOS authors have the option to publish the peer review history of their article (what does this mean?). If published, this will include your full peer review and any attached files.

Reviewer #1: No

Reviewer #2: No

Reviewer #3: No

---

## [Decision Letter · Decision Letter 2]

9 Feb 2025

Dear Dr. Tune,

We are pleased to inform you that your manuscript 'Nanometer scale difference in myofilament lattice structure of muscle alters muscle function in a spatially explicit model' has been provisionally accepted for publication in PLOS Computational Biology.

Best regards,

Daniel A Beard

Section Editor

PLOS Computational Biology

Daniel Beard

Section Editor

PLOS Computational Biology

Reviewer's Responses to Questions

**Comments to the Authors:**

Reviewer #1: All of my comments have been addressed.

Reviewer #2: I think the paper is publication ready.

Reviewer #3: Thank you for updating the revised manuscript and addressing my questions.

**Have the authors made all data and (if applicable) computational code underlying the findings in their manuscript fully available?**

Reviewer #1: None

Reviewer #2: Yes

Reviewer #3: Yes

PLOS authors have the option to publish the peer review history of their article (what does this mean?). If published, this will include your full peer review and any attached files.

Reviewer #1: No

Reviewer #2: No

Reviewer #3: No

---

## [Editor Report · Acceptance letter]

PCOMPBIOL-D-24-00909R2

Nanometer scale difference in myofilament lattice structure of muscle alters muscle function in a spatially explicit model

Dear Dr Tune,

I am pleased to inform you that your manuscript has been formally accepted for publication in PLOS Computational Biology. Your manuscript is now with our production department and you will be notified of the publication date in due course.

With kind regards,

Lilla Horvath
